# Isolation and Characterization of *Mycoplasma ovipneumoniae* Infecting Goats with Pneumonia in Anhui Province, China

**DOI:** 10.3390/life14020218

**Published:** 2024-02-02

**Authors:** Jiahong Chen, Shijia Wang, Dong Dong, Zijun Zhang, Yafeng Huang, Yong Zhang

**Affiliations:** 1College of Veterinary Medicine, Gansu Agricultural University, Lanzhou 730070, China; chenjiahong@ahau.edu.cn; 2College of Animal Science and Technology, Anhui Agricultural University, Hefei 230036, China; wangsj2024@ahau.edu.cn (S.W.); zhangzijun@ahau.edu.cn (Z.Z.); huangyafeng@ahau.edu.cn (Y.H.); 3Center of Agriculture Technology Cooperation and Promotion of Dingyuan County, Dingyuan 233200, China; dongdong@ahau.edu.cn

**Keywords:** goats, pleuropneumonia, *Mycoplasma*, amplicon sequencing

## Abstract

*Mycoplasma ovipneumoniae* (*M. ovipneumoniae*) causes a fatal infection in goats, leading to significant economic losses in the small-ruminant industry worldwide. The present study aimed to characterize the strains of *M. ovipneumoniae* infecting goats with pneumonia in Anhui Province, China. From November 2021 to January 2023, among 20 flocks, a total of 1320 samples (600 samples of unvaccinated blood, 400 nasal swabs, 200 samples of pleural fluid, and 120 samples of lung tissue) were obtained from goats with typical signs of pneumonia, such as a low growth rate, appetite suppression, increased temperature, discharge from the nose, and a cough. Necropsied goats showed increased pleural fluid, fibrinous pleuropneumonia, and attached localized pleural adhesions. *M. ovipneumoniae* isolated from the samples were subjected to an indirect hemagglutination test (IHA), PCR amplicon sequencing, phylogenetic analysis, and biochemical identification tests. The overall positivity rate of *M. ovipneumoniae* was 27.50%. *Mycoplasmas* were obtained from 80 (20.0%) nasal swabs, 21 (10.5%) pleural fluid samples, and 15 (12.5%) lung samples. PCR amplicon (288 bp) sequencing identified eight strains of *M. ovipneumoniae*. In a phylogenetic tree, the isolated strains were homologous to the standard strain *M. ovipneumoniae* Y-98 and most similar to *M. ovipneumoniae* FJ-SM. Local strains of *M. ovipneumoniae* were isolated from goats in Anhui province. The identified genomic features and population structure will promote further study of *M. ovipneumoniae* pathogenesis and could form the basis for vaccine and therapy development.

## 1. Introduction

The livestock sector, particularly the small-ruminant industry, plays a significant socioeconomic role in numerous countries globally, as it efficiently converts human-inedible forage into high-value meat that meets the discerning demands of today’s consumers in terms of quality and sensory attributes [1]. According to statistics, the global population of goats and sheep steadily increased from 2.05 billion to 2.39 billion between 2011 and 2021, with China accounting for more than 13.35% [2]. However, the small-ruminant industry has also encountered significant economic losses due to elevated mortality rates, reduced carcass values, as well as costly prevention and treatment associated with respiratory diseases in sheep and goats [3]. In 1963, the first isolation of *M. ovipneumoniae* from the lung of a diseased sheep was reported in Scotland [4], where it was considered a fatal respiratory disease, posing a direct threat to the small-ruminant industry [5,6]. The characteristics of *M. ovipneumoniae* infection are a low growth rate, appetite suppression, discharge from the nose, pyrexia, and dyspnea [7]. In the USA, in 2001, among 435 sheep farms tested, 88% were positive for *M. ovipneumoniae* [8]. From 2007 to 2019, 16.4% of isolated *Mycoplasma* spp. from small ruminants in France were *M. ovipneumoniae* [9]. From 2005 to 2019, more than 50% of *Mycoplasmas* isolated from small ruminants in England and Wales were *M. ovipneumoniae* [10]. In recent years, the incidence of ruminant *Mycoplasma* pneumonia in China has increased.

Apparently healthy and diseased animals can test positive for *M. ovipneumoniae*, which is transmitted primarily via respiration after repeated close contact [11]. According to previous studies, *M. ovipneumoniae* has spread globally and has been reported in several regions, with an especially high incidence in lambs [8,9,12,13,14,15,16], and the *M. ovipneumoniae* strains isolated from infected flocks differ worldwide [17]. *M. ovipneumoniae* is a localized epidemic, with an average duration of 15 days in winter and over 2 months in summer. Other predisposing factors contributing to the disease include high stocking densities and low ventilation rates for intensely bred lambs [18], transportation emergencies [7], different ecoclimatic factors, and seasonal variation. Based on the course of disease and clinical symptoms, it can be classified into three categories: peracute, acute, and chronic. Additionally, the immunogenicity and virulence of the different *M. ovipneumoniae* strains vary because of intra-species genetic diversity and antigenic heterogeneity [19,20,21]. In flocks, some animals can even be infected with multiple strains, leading to peracute diseases, such as bronchopneumonia, polymicrobial pneumonia, arthritis, keratoconjunctivitis, concomitant lung lesions, and lung abscesses [22]. Once infected, secondary infections, like *Mannheimia haemolytica* and *Influenza D virus*, can occur along with mixed infections involving *Mycoplasma arginini*, which ultimately result in the death of the affected animals [23,24]. The progression of the disease depends on the presence of variable virulent strains, as well as the host response and co-infections [11]. In addition, *M. ovipneumoniae* can infect various wild ruminants, including endangered species, such as bighorn sheep (*Ovis canadensis*) [25], musk ox (*Ovibos moschatus*) [26], porcupine caribou (*Rangifer tarandus granti*) [27], and white-tailed deer (*Odocoileus virginianus*) [28]. Although many relevant studies have been conducted to date, few studies have reported on *M. ovipneumoniae* in the southern region of China. Considering the significance and widespread damage caused by M. ovipneumoniae, we undertook this study to characterize the strains of *M. ovipneumoniae* infecting goats with pneumonia in Anhui Province, China.

Huanghuai goats (*Capra hircus*) are an artificially cultivated variety from China that have gained a long-standing reputation in the international market for their exceptional adaptability, high fecundity, tolerance to roughage, and superior skin quality, and are mainly distributed in the Anhui, Henan, and Jiangsu Regions of China. However, with the increase in large-scale and intensive breeding practices in recent years, *M. ovipneumoniae* has emerged as one of the primary diseases affecting Huanghuai goat breeds. In this study, indirect hemagglutination (IHA), histopathology, pathogen culture, and polymerase chain reaction (PCR) were used to isolate and identify naturally infected samples, with the aim of understanding the infection situation of *M. ovipneumoniae* in Anhui province and the biological characteristics of the epidemic strains. Our results might lay a foundation to explore *M. ovipneumoniae* pathogenesis and propose prevention and control measures.

## 2. Materials and Methods

### 2.1. Region of Study and the Animals Used

Twenty goat flocks from Anhui Province were selected for this study (from November 2021 to January 2023). Female and male goats with a history and symptoms of respiratory infection, thinness, pyrexia, cough, and discharge from the nose were selected. The flocks comprised 200–3000 goats. The ethics committee of Anhui Agricultural University approved the animal procedures under permit No. AHAUB2022009.

### 2.2. Animals and Sampling

Samples (n = 1320, comprising 600 samples of unvaccinated blood, 400 nasal swabs, 200 samples of pleural fluid, and 120 samples of lung tissue) were collected from goats showing typical symptoms of *M. ovipneumoniae* infection. Blood samples were collected from 270 ewes and 330 rams, and 131 diseased goats were dissected. Isolated serum was screened for *M. ovipneumoniae* positivity using an indirect hemagglutination test kit (LSBIO, Lanzhou, China). Biochemical tests were carried out using microreaction tubes (Hangzhou Microbial Reagent Co., Ltd., Hangzhou, China). The external nares were cleaned properly, followed by the collection of nasal swabs, which were placed in broth culture. At necropsy, lung samples were collected aseptically from the diseased–healthy junction, and sterile syringes were used to collect pleural fluids. All samples were placed in an icebox (4 °C) for transport to the laboratory.

### 2.3. Histopathology

The fixed goat lung fragments were removed from Bouin’s solution, followed by washing with phosphate-buffered saline (PBS). Subsequently, the fragments were dehydrated using a gradient of ethanol, hyalinized in xylene, followed by paraffin wax embedding. Next, the embedded samples were sectioned at 6 µm and subjected to hematoxylin and eosin (H&E) staining using standard protocols. A DM750 microscope (Leica, Nussloch, Germany) was used to capture images of the stained sections.

### 2.4. Isolation and Identification of M. ovipneumoniae

*Mycoplasmas* in the samples were isolated from nasal swabs through cultivation on pleuro-pneumonia-like organism (PPLO) agar and broth using previously described methods [29,30]. Nasal swabs and pleural fluid samples were cultivated in 15 mL conical tubes containing PPLO broth (21 g/L; Qingdao Hopebio Biotechnology Co., Ltd., Qingdao, China) supplemented with heat-inactivated horse serum (200 mL/L; Gibco, Grand Island, NY, USA), 25% fresh yeast extract (100 mL/L; Oxoid, Hampshire, UK), polymyxin B sulfate (3 units/mL; Shanghai Yuanye Biotechnology Co., Ltd., Shanghai, China), penicillin G (200,000 IU/L; Shanghai Macklin Biotechnology Co., Ltd., Shanghai, China), D-glucose (2 g/L), pyruvic acid sodium salt (2 g/L), and 0.4% phenol red (0.18 mL; Beijing Solarbio Technology Co., Ltd., Beijing, China). The lung tissues at the boundary between diseased and healthy areas were cut into soybean-sized pieces, and then, ground. Pleural fluid and minced tissue suspensions (following vigorous shaking) were serially diluted in PPLO liquid medium 10-fold to 10^−3^, filtered using a 0.45 µm ministart filter (Millipore Corporation, Billerica, MA, USA) and incubated. The liquid cultures were examined daily over 1 week for evidence of growth (color change indicating altered pH and the appearance of flocculation). The liquid media in tubes showing growth were then subcultured on PPLO agar, containing PPLO agar (35 g/L; Qingdao Hopebio Biotechnology) supplement with heat-inactivated horse serum (200 mL/L; Gibco), 25% fresh yeast extract (100 mL/L; Oxoid), polymyxin B sulfate (3 units/mL; Shanghai Yuanye Biotechnology), penicillin G (200,000 IU/L; Shanghai Macklin Biotechnology), D-glucose (2 g/L), pyruvic acid sodium salt (2 g/L), and 0.4% phenol red (5 mL; Beijing Solarbio Technology). Plates were inspected every day under a stereo-microscope to check for the presence of characteristic *Mycoplasma* colonies. A small block of agar containing an isolated *Mycoplasma* colony was added to PPLO broth medium and incubated for 3 to 5 days at 37 °C to purify and clone the primary culture. Giemsa staining (Hopebio) and Dienes staining (Solarbio, Beijing, China) were used to permissively identify the isolates according to the morphology of their colonies.

### 2.5. Extraction of DNA and Amplification by PCR

The PPLO broth with *M. ovipneumoniae* grown to a logarithmic stage was transferred to a 1.5 mL centrifuge tube, centrifuged at 150,000 r for 10 min, to collect the cells, and DNA was then extracted using a TIANamp Bacteria DNA Kit (TIANGEN, Beijing, China) following the supplier’s guidelines. A NanoDrop 2000 UV Spectrophotometer (Thermo Fisher Scientific, Waltham, MA, USA) and 1% agarose gel electrophoresis were used to evaluate the DNA concentration and integrity, respectively.

PCR was performed using the *Mycoplasma mycoides cluster*-specific primers RNA5 (5′-AGAGTTTGATCCTGGGCTCAGGA-3′) and MGSO (5′-TGCACCATCTGTCACTCTGTTAACCTC-3′), yielding an amplicon of 1021 bp from the 16S rRNA gene [16]. An amplicon of 288 bp was produced by PCR using the *M. ovipneumoniae*-specific primers Movi-F (5′-AGCGTCTCACATTTTCGCAC-3′) and Movi-R (5′-CAAGCAAATCCCGAACCCTG-3′) [31]. The PCR reaction (25 µL) comprised 12.5 µL of Premix Taq™ (Takara, Shiga, Japan), 1 µL of 10 pmol/µL forward and reverse primers, 1 µL of genomic DNA template, and 9.5 µL of ddH_2_O. Negative and positive controls were set up at the same time, in which ddH_2_O was the negative control and purified DNA of *M. ovipneumoniae* Y98 was the positive control (kindly donated by Prof. Yuefeng Chu from Lanzhou Veterinary Research Institute (CAAS)). An I cycler PCR machine (BIO-RAD, Hercules, CA, USA) was used to carry out the PCR reactions, with the following thermo-cycling conditions: initial denaturation for 2 min at 98 °C, followed by 35 cycles of 94 °C for 10 s, 64.5 °C for 15 s, and 72 °C for 15 s and a final 72 °C extension step for 10min. The amplicons were subjected to 1% agarose gel electrophoresis incorporating ethidium bromide and analyzed using an ultraviolet gel documentation system (BIO-RAD). The PCR products were submitted to a biological company for sequencing (Shanghai Sangon Co., Ltd., Shanghai, China). The neighbor-joining method in the MEGA11.0 was used to generate a phylogenetic tree (using 1000 bootstrap replicates). Table 1 shows the details of the reference strains used in the phylogenetic tree [32].

### 2.6. Biochemical Characteristic Tests

The biochemical characteristic tests inclued Digitonin sensitivity test, Glucose fermentation test, Arginine hydrolysis test and Urea hydrolysis test. The tests were carried out strictly following the instructions in the user manual of each biochemical identification tube.

## 3. Results

### 3.1. Symptoms

The manifestations of *M. ovipneumoniae* infection in goats were coughing, fever, labored breathing, mucoid nasal discharge, and gradual emaciation, with a usual temperature of 41°C, which occasionally reached 42 °C. As shown in Table 2, among the 600 serum samples collected from goats in Anhui Province, the positivity rate of *M. ovipneumoniae* was 27.50% according to the IHA test. The positivity rate was 40.5% in lambs, 28.5% in fattened lambs, and 13.5% in adult goats. The infection rates in spring, summer, autumn, and winter were 36.7%, 34.7%, 24.0%, and 27.5%, respectively. Peracute and acute cases were common in spring and winter, and chronic cases were common in summer.

### 3.2. Pathological Manifestation of M. ovipneumoniae Infection

Necropsy revealed a yellowish fluid in the pleural cavity and hepatization of the lungs in the lung parenchyma, whose cross-sections showed marble-like lesions (Figure 1). We observed enlarged lymph nodes with hemorrhagic spots, and varying degrees of enlargement of the liver, spleen, kidneys, and gallbladder. Histopathological examination of the lungs showed marked thickening of the alveolar walls and widening of the interstitial spaces, accompanied by high granulocytic infiltration; lymphocytes were focally distributed around the bronchi and blood vessels (Figure 2).

### 3.3. Frequency of M. ovipneumoniae Infection among the Goats

The isolation data of *M. ovipneumoniae* are shown in Table 3. The probability of *M. ovipneumoniae* isolation from nasal swabs was 20%, from pleural effusion was 10.5%, and from lung tissue was 12.5%. The PPLO broth became turbid within 3 to 7 d. Under the microscope, within 2 to 3 d, the colonies on PPLO agar were round, without a central umbilicus. Stereoscopic microscopy revealed blue Dienes staining of the colonies, indicating that they were *Mycoplasmas*. Oil microscopy showed lavender Giemsa staining of the cell body, with a predominance of spherical and filamentous forms (Figure 3).

### 3.4. Molecular Characterization

The *Mycoplasma mycoides cluster* 16S rRNA gene (1021 bp) was amplified from 91 *Mycoplasma* isolates. Further PCR analysis of the *Mycoplasma* isolates using *M. ovipneumoniae*-specific primers confirmed eight strains as *M. ovipneumoniae* (288 bp) (Figure 4). The PCR products of the target bands were sequenced and analyzed by BLAST comparison, which showed 99.05% similarity with the standard strain *M. ovipneumoniae* Y-98 strain in GenBank, and the phylogenetic tree showed that they had the highest similarity to *M. ovipneumoniae* FJ-SM (GenBank accession number: KU870650.1) (Figure 5).

### 3.5. Biochemical Test Results

The biochemical identification results of the eight strains were in line with the biochemical characteristics of the *M. ovipneumoniae* standard strain Y-98, i.e., they were sensitive to digitalin; able to ferment glucose; reduced tetrazolium chloride; and did not hydrolyze arginine and urea. The results are presented in Table 4.

## 4. Discussion

To the best of our knowledge, this was the first investigation of *M. ovipneumoniae* infections in Huanghuai goats within Anhui province, China. *M. ovipneumoniae* poses a common and costly threat to the goat meat industry by inducing low growth rates and reduced productivity among infected animals, thereby exerting significant adverse effects on economic stability [8,32]. *M. ovipneumoniae* infection leads to diseases known as *Mycoplasma* pneumonia, non-progressive (atypical) pneumonia of sheep, chronic bronchopneumonia, and proliferating exudative pneumonia. Infected goats can transmit pathogens to other goats through direct contact and aerosol transmission via coughing, sneezing, and nasal secretions [8,33,34,35,36]. In the present study, Huanghuai goats were used as an experimental model to investigate *M. ovipneumoniae* infections. The clinical symptoms and postmortem lesions of the goats in this study were similar to those described previously for *M. ovipneumoniae* infections. The clinical symptoms presented as both acute and chronic manifestations, with acute infection characterized by a sudden rise in body temperature accompanied by a transient cough and increased viscosity of nasal discharge. Subsequently, persistent high fever ensues, along with pronounced anorexia, respiratory distress, purulent ocular discharge, and weight loss. Chronic symptoms were relatively mild and more frequently observed during the summer months; they typically include intermittent episodes of coughing, followed by gradual weight loss [7]. The lung surface exhibited heterogeneous characteristics, with areas displaying varying sizes, indicative of hepatization. Externally, the section appeared reddish with a medium gray hue resembling marble, which is suggestive of typical interstitial pneumonia. Dilatation of the bronchial tubes and the formation of thrombi in blood vessels were observed. The pleura was covered in rough yellowish-white fibrin, while the chest displayed a yellowish coloration and contained a significant amount of fluid that tended to solidify into a jelly-like consistency upon exposure to air. Adhesion between the lungs and pericardium as well as enlargement of the pulmonary lymph nodes were noted. Furthermore, swelling was evident in the liver, gallbladder, spleen, and kidneys. The above symptoms and morphologies are consistent with those reported previously for *M. ovipneumoniae* infections [37,38,39,40].

The indirect hemagglutination test offers the advantages of speed, high sensitivity, and efficacy in detecting large quantities of blood samples. However, cross-reactivity caused by common antigens between *Mycoplasma* species might lead to false positive or negative results [41]. In this study, we utilized the indirect hemagglutination test to detect *M. ovipneumoniae* infections in domestic goats. Our findings indicated that lambs are particularly susceptible, with a higher mortality rate around 30 days old, while fattening sheep at 8–10 months old also exhibited high susceptibility. No age effects were observed in adult sheep infections. Furthermore, our research revealed that lambs are most vulnerable to infection in March, while August poses a greater risk for fattened goats, which is consistent with the results of previous research [42]. Recent reports from regions experiencing significant losses have frequently detected *M. ovipneumoniae* [9,15,16]. In China, from 2012 to 2014, indirect hemagglutination tests were employed to test 1679 blood samples across six regions of Xinjiang. The analysis revealed a positivity rate of 17.75% of the total samples tested. It was suspected that inter-provincial transmission had facilitated the epidemic’s spread [43]. From 2012 to 2013, an indirect hemagglutination test detected an *M. ovipneumoniae*-positivity rate of 31.7% in goats inhabiting tropical areas of China [44]. Its prolonged presence in animals without evident clinical symptoms at disease onset has made *Mycoplasma* challenging to detect and diagnose promptly in China. This delay leads to more severe conditions and increased mortality rates when symptoms become apparent. Therefore, our study underscores the necessity for further research on local *M. ovipneumoniae*-related information.

Although fast and reliable kits are available to detect *M. ovipneumoniae*, *Mycoplasma* culture remains critical because it serves as an important basis for strain identification, susceptibility testing, drug therapy, and vaccine development [29,45]. The commonly used media for culturing *M. ovipneumoniae* are primarily PPLO and Hayflick’s medium with the addition of polymyxin to inhibit walled bacteria, which leads to better culture results [46,47]. Additionally, placing the sample in a medium at 37 °C in a 5% CO_2_ atmosphere with 98% humidity significantly enhances the growth rate, because higher temperatures and lower relative humidity favor the presence of *Mycoplasma* species [14,40]. It was observed that sediment began to appear at the bottom of the tube on day three when using PPLO broth medium for culture. Furthermore, by transferring it to PPLO agar on day three, we could visually observe changes in phenol red indicator coloration. By days 5 to 6, the cultures had reached the late growth stage, which aligns with the findings of previous research [29]. These observations indicate that the effective culturing of most *M. ovipneumoniae* strains is crucial within the first three days.

The diagnosis of *Mycoplasma* infection typically involves a combination of polymerase chain reaction (PCR) and serology, which can be rapidly applied to clinical samples for gaining an overall understanding of the epidemic situation. PCR is a more reliable method for *Mycoplasma* growth assays, even in the presence of bacterial contamination [11,48]. Screening isolated genomic DNA using the 16S rRNA gene successfully produced the expected 288 bp amplicon, confirming that all the isolates belonged to the *Mycoplasma mycoides cluster*. The subsequent sequencing of purified PCR amplicons identified eight strains as *M. ovipneumoniae*. Furthermore, out of 720 clinical samples (nasal swabs: 20.0%, pleural fluid: 10.5%, and lung samples: 12.5%), a total of 116 *M. ovipneumoniae* isolates were recovered, with a positivity rate of 16.1%. The high isolation rate from nasal swabs might be attributed to respiratory secretion being the primary route through which diseased and infected goats eliminate *M. ovipneumoniae*. The molecular pathogenesis of *M. ovipneumoniae* has been determined. Upon entry into the respiratory tract, *M. ovipneumoniae* adheres to the surface of tracheal cilia and alveolar cells, resulting in diffuse panbronchiolitis, which rapidly extends to the surrounding interstitium. Colonization by *M. ovipneumoniae* is facilitated through interactions between its expressed adhesin proteins and sulfated glycolipids or sialoglycoprotein molecules present on the host respiratory epithelium, ultimately leading to community-acquired pneumonia. Subsequently, dissemination occurs via lymphatic vessels and blood vessels, reaching the lungs [49,50]. Previous studies have demonstrated that the trachea of diseased sheep harbored the highest loads of *M. ovipneumoniae* nucleic acids, followed by the lungs. Relatively lower loads were detected in the kidneys, spleen, and liver [51,52]. Importantly, an increased detection rate of *M. ovipneumoniae* could serve as a crucial wake-up call for the industry. Furthermore, the high prevalence of *M. ovipneumoniae* among tested goats in this region might be attributed to various factors, including temperature, humidity, stress levels, and epidemic strains. Therefore, greater attention should be devoted to the implementation of proposed preventive measures. Given the substantial yet uncertain impact of *M. ovipneumoniae* on the small-ruminant industry, it is imperative to incorporate prevention and control strategies into daily feeding management practices. The endemic nature of this disease, coupled with its airborne transmission and host susceptibility in cold, wet, and crowded conditions, necessitates precautionary actions. Inadequate nutrition during spring and winter exacerbates animals’ vulnerability to respiratory illnesses, emphasizing the significance of refraining from introducing new animals during these seasons, quarantining exotic species prior to their integration into herds, isolating potentially ill individuals for quarantine purposes, and continuously monitoring herd health status. Phylogenetic trees depict the evolutionary relationships among species that are postulated to share a common ancestor and constitute a single lineage on the tree of life [53]. The phylogenetic analysis in this study revealed that the isolate exhibited distant relatedness to *Escherichia coli* (GenBank accession number: X80725.1) and displayed the highest sequence similarity (99.47%) between BZ-10 and *M. ovipneumoniae* FJ-SM, aligning well with the conventional classification of *Mycoplasmas*. Furthermore, the BLAST results demonstrated 99.05% sequence identity between the isolated strain and the reference strain *M. ovipneumoniae* Y-98. Currently, the limited availability of commercial vaccines for *M. ovipneumoniae* can be attributed to insufficient disease awareness and attention, as well as cost-saving practices in drug usage among farmers. Moreover, vaccines demonstrate restricted cross-protection against isolates from diverse regions. The limited success of reported experimental vaccines might arise from both the high variability of *M. ovipneumoniae* and its immune evasion strategies [46,54]. Herein, the results of the biochemical tests were consistent with those reported in the literature [55].

The currently used inactivated vaccine for goat *Mycoplasma* pneumonia in China is derived from local endemic strains *M. ovipneumoniae* MOGH3-3 and *Mycoplasma mycoides* subsp. *capri* M87-1. It exhibits a protection rate of over 90% and provides immune protection for at least 10 months, making it suitable for sheep and goats. However, the control of *M. ovipneumoniae* infection primarily relies on antimicrobial therapy because of its inherent resistance to cell wall-targeting antimicrobial drugs such as β-lactams and glycopeptide antibiotics. Oxytetracycline and macrolide antibiotics can alleviate clinical symptoms, but fail to eradicate the infection [56]. Notably, fluoroquinolones, tetracyclines, and macrolides have demonstrated efficacy in vitro to treat *Mycoplasma pneumoniae* [57]. Considering the limited annotation of virulence factors for *M. ovipneumoniae* from previous studies, future research should employ transcriptomic and metabolomic methods to accurately characterize unknown virulence genes, determine their prevalence, and elucidate their mechanisms of action to develop an optimal treatment strategy.

## 5. Conclusions

This study employed an indirect hemagglutination test to investigate the prevalence of *M. ovipneumoniae* infection in goats in Anhui Province. Pathological sections were prepared from the dissected lungs of Huanghuai goats, while isolates obtained from nasal swabs, pleural effusions, and lung tissues were subjected to phylogenetic analysis. The results revealed a regional distribution of *M. ovipneumoniae* infection within Anhui Province, with acute or subacute onset predominantly occurring during winter and spring; however, chronic cases primarily manifested during summer months and had a significant impact on the health status of lambs. Notably, this study successfully isolated and purified eight local strains of *M. ovipneumoniae* from naturally infected goats in Anhui Province for the first time within this region. These strains exhibited a high degree of genetic similarity (99.05%) with the standard strain *M. ovipneumoniae* Y-98, as documented in GenBank, while phylogenetic analysis revealed that they shared the closest similarity to *M. ovipneumoniae* FJ-SM. These findings lay the foundation for conducting whole-genome sequencing studies and further research. Our findings also have significant implications for molecular-based diagnosis and vaccine development, aiming at targeted treatment and prevention measures to effectively mitigate disease outbreaks among goats.

## Figures and Tables

**Figure 1 life-14-00218-f001:**
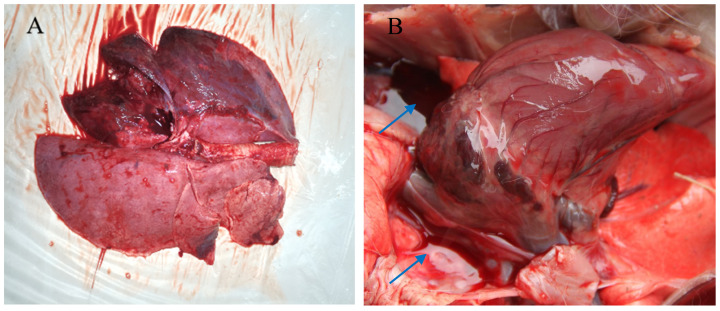
Gross morphology of the lungs from a goat infected with *Mycoplasma ovipneumoniae.* (**A**) Irregular red consolidations; (**B**) pleural effusion; (**C**) trachea filled with mucus; (**D**) adhesions; (**E**) hepatization of the lungs; (**F**) cross-sections of hepatization. The blue arrows indicate conspicuous lesions.

**Figure 2 life-14-00218-f002:**
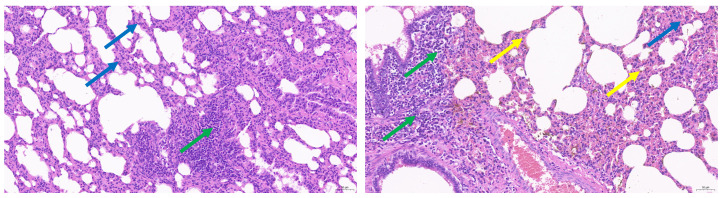
Hematoxylin and eosin (HE) staining analysis (200×) of the lungs from a goat infected with *Mycoplasma ovipneumoniae*. Blue arrows: granulocyte infiltration; green arrows: lymphocytic infiltration; yellow arrows: alveolar wall capillary congestion.

**Figure 3 life-14-00218-f003:**
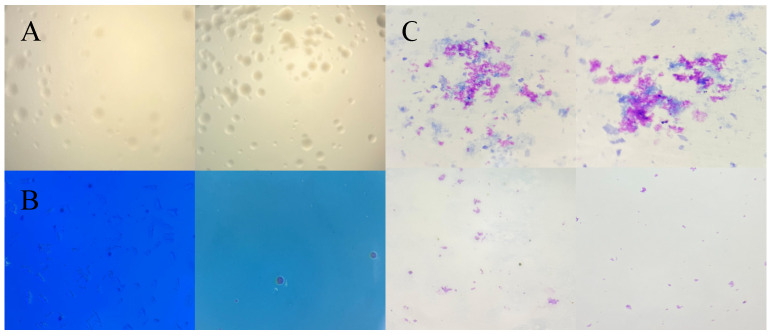
Colonies of *Mycoplasma ovipneumoniae* on PPLO agar, characterized as round without a central umbilicus. (**A**) Purified colonies of an isolated strain (400×). (**B**) Dienes staining of purified colonies (10×). (**C**) Giemsa staining of purified colonies (400×).

**Figure 4 life-14-00218-f004:**
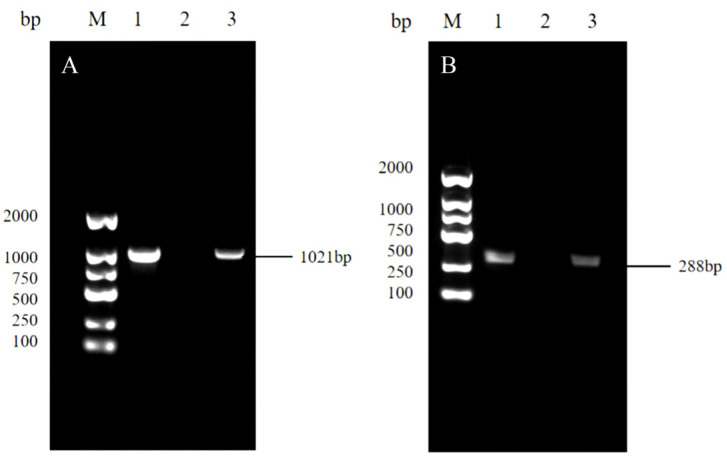
Partial PCR amplicons obtained using *Mycoplasma mycoides cluster*-specific primers (**A**) and *Mycoplasma ovipneumoniae-*specific primers (**B**). Lane M: DNA ladder marker (2000 bp); Lane 1: *Mycoplasma ovipneumoniae*-positive control; Lane 2: negative control; and Lane 3: *Mycoplasma ovipneumoniae* field isolates.

**Figure 5 life-14-00218-f005:**
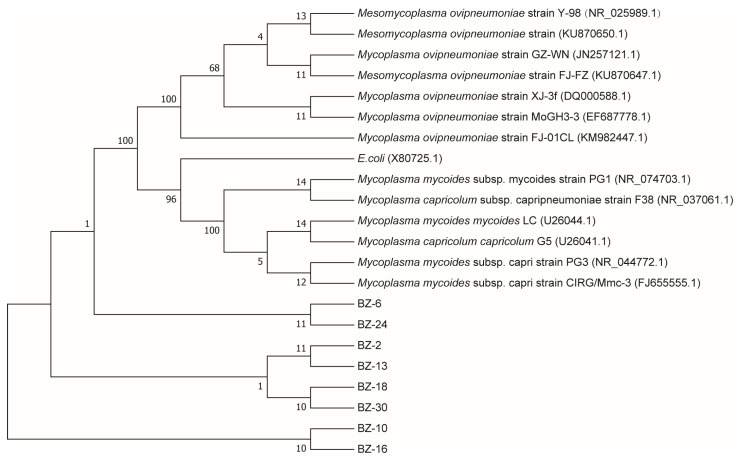
Phylogenetic tree analysis of *M. ovipneumoniae* based on partial sequences. Each sequence was downloaded from the NCBI database and labeled with the species names and GenBank accession numbers. BZ represents the strain of *M. ovipneumoniae* isolated from goats in the present study. The tree was generated using the neighbor-joining method (using 1000 bootstrap replicates).

**Table 1 life-14-00218-t001:** Species used for the phylogenetic analysis.

Number	Species	GenBank Accession Numbers
1	*Mesomycoplasma ovipneumoniae* strain Y-98	NR025989.1
2	*Mesomycoplasma ovipneumoniaee* strain FJ-SM	KU870650.1
3	*Mesomycoplasma ovipneumoniaee* strain FJ-FZ	KU870647.1
4	*Mycoplasma ovipneumoniae* strain MoGH3-3	EF687778.1
5	*Mycoplasma ovipneumoniae* strain GZ-WN	JN257121.1
6	*Mycoplasma ovipneumoniae* strain XJ-3f	DQ000588.1
7	*Mycoplasma ovipneumoniae* strain FJ-01CL	KM982447.1
8	*Mycoplasma capricolum capricolum* strain G5	U26041.1
9	*Mycoplasma capricolum* subsp. *capripneumoniae* strain F38	NR037061.1
10	*Mycoplasma mycoides* subsp. *Mycoides* strain PG1	NR074703.1
11	*Mycoplasma mycoides mycoides* LC type Y-goat	U26044.1
12	*Mycoplasma mycoides* subsp. *capri* strain PG3	NR044772.1
13	*Mycoplasma mycoides* subsp. *capri* strain CIRG/Mmc-3	FJ655555.1
14	*Escherichia coli* (ATCC 11775T)	X80725.1
15	BZ-2 (*Mesomycoplasma ovipneumoniae* strain AH2201)	OR964948.1
16	BZ-6 (*Mesomycoplasma ovipneumoniae* strain AH2202)	OR964949.1
17	BZ-10 (*Mesomycoplasma ovipneumoniae* strain AH2203)	OR964950.1
18	BZ-13 (*Mesomycoplasma ovipneumoniae* strain AH2204)	OR964951.1
19	BZ-16 (*Mesomycoplasma ovipneumoniae* strain AH2205)	OR964952.1
20	BZ-18 (*Mesomycoplasma ovipneumoniae* strain AH2206)	OR964953.1
21	BZ-24 (*Mesomycoplasma ovipneumoniae* strain AH2207)	OR964954.1
22	BZ-30 (*Mesomycoplasma ovipneumoniae* strain AH2208)	OR964955.1

**Table 2 life-14-00218-t002:** The *M. ovipneumoniae*-positivity rate data for different growth stages and different seasons.

Growth Stage Season	Lambs		Fattening Lambs		Adult Goats		Total	
	Positive /Total	Prevalence (%)	Positive /Total	Prevalence (%)	Positive /Total	Prevalence (%)	Positive /Total	Prevalence (%)
Spring	32/50	64.0	15/50	30.0	8/50	16.0	55/150	36.7
Summer	23/50	46.0	24/50	48.0	5/50	10.0	52/150	34.7
Autumn	9/50	18.0	10/50	20.0	3/50	6.0	22/150	14.7
Winter	17/50	34.0	8/50	16.0	11/50	22.0	36/150	24.0
Total	81/200	40.5	57/200	28.5	27/200	13.5	165/600	27.5

**Table 3 life-14-00218-t003:** Isolation rate of *M. ovipneumoniae* from different samples.

Sample Type	Sample Count	Positive	Detection Rate (%)
Nasal swabs	400	80	20.0
Pleural fluid	200	21	10.5
Lung tissue samples	120	15	12.5

**Table 4 life-14-00218-t004:** Biochemical identification of the isolated strains.

Biochemical Test	Isolated Strains								
	Y98	BZ-2	BZ-6	BZ-10	BZ-13	BZ-16	BZ-18	BZ-24	BZ-30
Digitonin sensitivity test	+	+	+	+	+	+	+	+	+
Glucose fermentation test	+	+	+	+	+	+	+	+	+
Arginine hydrolysis test	−	−	−	−	−	−	−	−	−
Urea hydrolysis test	−	−	−	−	−	−	−	−	−

Note: ”+” Positive, ”−” Negative.

## Data Availability

The sequencing data reported in this study were deposited in GenBank, available online at https://www.ncbi.nlm.nih.gov/genbank/ (accessed on 20 December 2023 (OR964948, OR964949, OR964950, OR964951, OR964952, OR964953, OR964954, and OR964955)).

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
