# Peer review of "Isolation and Characterization of Mycoplasma ovipneumoniae Infecting Goats with Pneumonia in Anhui Province, China"

_life, 2024, doi:10.3390/life14020218_

Round 1

Reviewer 1 Report

Comments and Suggestions for Authors

Comment to "Isolation and Characterization of Mycoplasma ovipneumoniae infecting Goats with Pneumonia in Anhui Province, China"

The article is important, which revealed the local strain of M. ovipneumoniae isolated from goats in Anhui province. one question is listed below:

1. The authors had better make a table to compare the difference between the present Mycoplasma ovipneumoniae with the previous Mycoplasma ovipneumoniae, such as Hematoxylin and eosin changes.

Author Response

Dear Editors and Reviewers:

Thank you for your letter and for the reviewers’ comments concerning our manuscript (ID: life-2811880). Those comments are all valuable and very helpful for revising and improving our paper, as well as providing important guiding significance to our research. We have studied the comments carefully and have made corrections that we hope meet with approval. The revised portions are marked using the "Track Changes" function in this paper. The main corrections in the paper and the responds to the reviewer’s comments are as follows:

Responds to the reviewer’s comments and suggestions:

  1. The authors had better make a table to compare the difference between the present Mycoplasma ovipneumoniaewith the previous Mycoplasma ovipneumoniae, such as Hematoxylin and eosin changes.

Response: As suggested by the reviewer, we have added a table in the revised manuscript. The main content is the comparison of biochemical test results between the eight strains and the standard strain, including Digitoxin sensitive test, Glucose fermentation test, Arginine hydrolysis test and Urea hydrolysis test. The biochemical identification results of the eight strains were in line with the biochemical characteristics of M. ovipneumoniae standard strain Y-98, the details can be found in Table 4. At present, only the analysis of pathological sections in this study mentions hematoxylin and eosin staining. I will check the relevant information carefully later. Additional minor revisions were made to the manuscript using track changes to improve English and readability based on suggestions from Elixigen CO. The comments you provided are greatly appreciated. The revisions we have made are expected to receive recognition.

Table 4. Biochemical identification of the isolated strains

Biochemical test

Isolated strains

Y98

BZ-2

BZ-6

BZ-10

BZ-13

BZ-16

BZ-18

BZ-24

BZ-30

Digitoxin sensitive test

+

+

+

+

+

+

+

+

+

Glucose fermentation test

+

+

+

+

+

+

+

+

+

Arginine hydrolysis test

Urea hydrolysis test

We tried our best to improve the manuscript and made some changes. Additional minor revisions were made to the manuscrip using track changes to improve English and readability based on suggestions from Elixigen CO. Special thanks for your good comments. We hope that the corrections made will meet with approval.

Reviewer 2 Report

Comments and Suggestions for Authors

The paper reviewed here concerns the pathology and characterisation of Mycoplasma ovipneumoniae strains isolated from Huanghuai goats in the southern region of China.

In my opinion, certain parts require language editing, because certain terms are used incorrectly making some sentences quite confusing.

Below are my notes referring to particular Line numbers and Figures:

56 – “peracute” is a more appropriate term here than “most acute”

91- how many animals were necropsied for the sample collection and the gross changes evaluation?

137- PPLO – instead of pplo

138 - add “centrifuged at” to the sentence

174 – “hepatisation of the lungs” is an appropriate term not “ hepatic changes”

175- “cross-sections” rather than “sections”

Figure 2. The quality of the photograph is too low to see any granulocytes- the blue arrows point to the widened interstitial spaces, so my advice is to change the description accordingly if higher magnification showing granulocytes is unavailable.

Figure 3 – The magnification should be in the description, because there is no scale bar on the photographs, especially in picture C showing different photographs.

203 – eight isolates or eight strains?

235- can you discuss whether the chronic symptoms were observed in the studied goats? The occurrence of the symptoms throughout the seasons, which were also recalled in “Conclusions” lines 347-348, were not mentioned in the “Results” section.

238 – as mentioned above, the term “hepatisation” should be used instead of “hepatic degeneration”

237-245 – The style of the section should be changed, because in the present form, the whole section is written like the description of the gross pathology in the results, instead of a comparison of the results with the cited literature.

247-248 – A citation accompanying this statement is required.

I think the authors should discuss how- or if- other aetiologic factors, especially other Mycoplasma spp. and other bacteria could contribute to the clinical and pathological changes observed in the studied population of goats.

Comments on the Quality of English Language

In my opinion, certain parts require language editing, because certain terms are used incorrectly making some sentences quite confusing.

Author Response

Dear Editors and Reviewers:

Thank you for your letter and for the reviewers’ comments concerning our manuscript (ID: life-2811880). Those comments are all valuable and very helpful for revising and improving our paper, as well as providing important guiding significance to our research. We have studied the comments carefully and have made corrections that we hope meet with approval. The revised portions are marked using the "Track Changes" function in this paper based on suggestions from Elixigen CO. The main corrections in the paper and the responds to the reviewer’s comments are as follows:

Responds to the reviewer’s comments and suggestions:

  1. L56 “ peracute” is a more appropriate term here than “most acute”

Response: Thank you for your suggestions, which are very important for the improvement of our manuscript. As suggested by the reviewer, we changed from “most acute” to “peracute” within line 61 of the revised manuscript.

  1. L91 how many animals were necropsied for the sample collection and the gross changes evaluation?

Response: Thank you for your suggestions, which are very important for the improvement of our manuscript. As suggested by the reviewer, we added “131 diseased goats were dissected” within lines 102 of the revised manuscript.

  1. L137 PPLO instead of pplo

Response: Thank you for your suggestions, which are very important for the improvement of our manuscript. As suggested by the reviewer, we edited this word in line 147 of the revised manuscript.

  1. L138 add “centrifuged at” to the sentence

Response: As suggested by the reviewer, we changed from “The pplo broth with M. ovipneumoniae growth to logarithmic stage was transferred to a 1.5 mL centrifuge tube at 150000 r for 10 min to collect sediment” to “The PPLO broth with M. ovipneumoniae grown to logarithmic stage was transferred to a 1.5 mL centrifuge tube, centrifuged at 150000 r for 10 min to collect the cells”.

  1. L174 “hepatisation of the lungs” is an appropriate term not “ hepatic changes”

Response: Thank you for your suggestions, which are very important for the improvement of our manuscript. As suggested by the reviewer, we edited these words in line 192 of the revised manuscript.

  1. L175 “cross-sections” rather than “sections”

Response: As suggested by the reviewer, we changed from “A necropsy revealed a yellowish fluid in the pleural cavity and hepatic changes in the lung parenchyma, whose sections showed marble-like lesions (Fig 1)” to “Necropsy revealed a yellowish fluid in the pleural cavity and hepatisation of the lungs in the lung parenchyma, whose cross-sections showed marble-like lesions (Fig. 1)”.

  1. Figure 2. The quality of the photograph is too low to see any granulocytes- the blue arrows point to the widened interstitial spaces, so my advice is to change the description accordingly if higher magnification showing granulocytes is unavailable.

Response: We agree with this statement. As suggested by the reviewer, The microscope images were reselected, the slices were reanalyzed, and the proportions were labeled in the diagram.

Fig. 2 Hematoxylin and eosin (HE) staining analysis(200×) of the lungs from a goat infected with Mycoplasma ovipneumoniae. Blue arrows: granulocyte infiltration; Green arrows: lymphocytic infiltration; Yellow arrows: alveolar wall capillary congestion.

  1. Figure 3 The magnification should be in the description, because there is no scale bar on the photographs, especially in picture C showing different photographs.

Response: Thank you for your suggestions, which are very important for the improvement of our manuscript. As suggested by the reviewer, we have enhanced the depiction of picture information and incorporated crucial details regarding microscope magnification in the description.

Fig 3. Colonies of Mycoplasma ovipneumoniae on PPLO agar, characterized as round without a central umbilicus. (A) Purified colonies of an isolated strain (400×). (B) Dienes staining of purified colonies (10×). (C) Giemsa staining of purified colonies (400×).

  1. L203 eight isolates or eight strains?

Response: The reviewer's meticulous examination is greatly appreciated. Our experiments resulted in the isolation of eight strains with different sequences, all of which have been uploaded to GenBank, available online at https://www.ncbi.nlm.nih.gov/genbank/ (accessed on 20 December 2023 (OR964948, OR964949, OR964950, OR964951, OR964952, OR964953, OR964954, andOR964955)). As suggested by the reviewer, we changed from “eight isolates” to “eight strains” within line 224 of the revised manuscript.

  1. L235 can you discuss whether the chronic symptoms were observed in the studied goats? The occurrence of the symptoms throughout the seasons, which were also recalled in “Conclusions” lines 347-348, were not mentioned in the “Results” section.

Response: We extend our sincere gratitude to the reviewers for their unwavering dedication. In Result 3.1, we demonstrate that M. ovipneumoniae infection manifests as cough, fever, dyspnea, mucoid nasal discharge, progressive wasting, typically at 41 °C and occasionally at 42 °C; these symptoms are indicative of a chronic onset. The discussion on lines 166-168 also addresses the presence of chronic symptoms. Additionally, we incorporated statistical findings on incidence rates during various seasons. The infection rates in spring, summer, autumn, and winter were 36.7%, 34.7%, 24.0%, and 27.5%, respectively.

  1. L238 as mentioned above, the term “hepatisation” should be used instead of “hepatic degeneration”

Response: Thank you for your suggestions, which are very important for the improvement of our manuscript. As suggested by the reviewer, the word has been changed in line 268 of the revised manuscript.

  1. L237-245 The style of the section should be changed, because in the present form, the whole section is written like the description of the gross pathology in the results, instead of a comparison of the results with the cited literature.

Response: Thank you for your suggestions, which are very important for the improvement of our manuscript. As suggested by the reviewer, we have made appropriate changes to add statements for comparison with other literature.

  1. L247-248 A citation accompanying this statement is required.

Response: Thank you for your suggestions, which are very important for the improvement of our manuscript. We extend our sincere gratitude to the reviewers for their unwavering patience and meticulous attention. As suggested by the reviewer, the citation has been added.

  1. I think the authors should discuss how- or if- other aetiologic factors, especially other Mycoplasma spp. and other bacteria could contribute to the clinical and pathological changes observed in the studied population of goats.

Response: We couldn't agree more with the reviewers. In lines 67-69 of the revised manuscript, we mentioned that M. ovipneumoniae can cause co-infection with Mannheimia haemolytica, Influenza D virus and Mycoplasma arginini, which eventually leads to the death of the animal. In future research, we will meticulously examine the information from Pathology.

We tried our best to improve the manuscript and made some changes. Additional minor revisions were made to the manuscrip using track changes to improve English and readability based on suggestions from Elixigen CO. Special thanks for your good comments. We hope that the corrections made will meet with approval.

Reviewer 3 Report

Comments and Suggestions for Authors

The manuscript "Isolation and Characterization of Mycoplasma ovipneumoniae infecting Goats with Pneumonia in Anhui Province, China" is an interesting study of great clinical significance. The literature is well selected. The text is enriched with photos which is especially important. The introduction is a proper introduction to the topic, the discussion is interestingly written.

Here are some comments:

- please clearly state the purpose of the study

- please include more information about the animals: age of the animals; how many females and how many males; what prophylaxis is used in the flocks

- the number of n is not correct: Samples (n = 1520, comprising 600 samples of unvaccinated blood, 400 nasal swabs, 200 samples of pleural fluid, and 120 samples of lung tissue). 600+400+200+120 = 1320

- all text should be written in the passive: "It was observed..." instead of "We also observed...".

- a valuable addition would be to add photos of more lesions of other organs in which lesions were found

- please describe the clinical manifestations in more detail; there is a lot of information in the discussion, but not much in the results that the researchers observed; if fever, how high...; what was the plan of the clinical study

- the volume number should be written in italics

Author Response

Dear Editors and Reviewers:

Thank you for your letter and for the reviewers’ comments concerning our manuscript (ID: life-2811880). Those comments are all valuable and very helpful for revising and improving our paper, as well as providing important guiding significance to our research. We have studied the comments carefully and have made corrections that we hope meet with approval. The revised portions are marked using the "Track Changes" function in this paper based on suggestions from Elixigen CO. The main corrections in the paper and the responds to the reviewer’s comments are as follows:

Responds to the reviewer’s comments and suggestions:

  1. Please clearly state the purpose of the study

Response: There has been no successful isolation of local strains in Anhui Province to date, which making it difficult to prevent and control Mycoplasma ovipneumoniae infection in the province. In this study, eight local strains were successfully isolated, which lays a foundation for disease diagnosis, vaccine development, and the study of its pathogenesis.

  1. Please include more information about the animals: age of the animals; how many females and how many males; what prophylaxis is used in the flocks

Response: As suggested by the reviewer, we added this information in lines 101-102 of the revised manuscript. The experiment involved the sampling of 270 ewes and 330 rams in total. The infection rates at various growth stages were delineated in the findings, as illustrated in Table 2. Blood specimens were obtained from animals that had not received vaccination.

Table 2. The M. ovipneumoniae positive rate data for different growth stages.

Growth         stage

Season

Lambs

Fattening lambs

Adult goats

Total

Positive

/Total

Prevalence

(%)

Positive

/Total

Prevalence

(%)

Positive

/Total

Prevalence

(%)

Positive

/Total

Prevalence

(%)

Spring

32/50

64.0

15/50

30.0

8/50

16.0

55/150

36.7

Summer

23/50

46.0

24/50

48.0

5/50

10.0

52/150

34.7

Autumn

9/50

18.0

10/50

20.0

3/50

6.0

22/150

14.7

Winter

17/50

34.0

8/50

16.0

11/50

22.0

36/150

24.0

Total

81/200

40.5

57/200

28.5

27/200

13.5

165/600

27.5

  1. The number of n is not correct: Samples (n = 1520, comprising 600 samples of unvaccinated blood, 400 nasal swabs, 200 samples of pleural fluid, and 120 samples of lung tissue). 600+400+200+120 = 1320

Response: The reviewer's meticulous correction is greatly appreciated. As suggested by the reviewer, we added changed this information in line 99 of the revised manuscript.

  1. All text should be written in the passive: "It was observed..." instead of "We also observed...".

Response: Thank you for your suggestions, which are very important for the improvement of our manuscript. As suggested by the reviewer, we changed from “We also observed...” to “It was observed...” within the revised manuscript.

  1. A valuable addition would be to add photos of more lesions of other organs in which lesions were found

Response: We really appreciate the reviewer's suggestion and agree with it. As suggested by the reviewer, the lesion photos captured during the collection process were incorporated into Figure 1.

Fig. 1 Gross morphology of the lungs from a goat infected with Mycoplasma ovipneumoniae. (A) Irregular red consolidations; (B) Pleural effusion; (C) Trachea filled with mucus; (D) Adhesions; (E) Hepatisation of the lungs; (F) Cross-sections of hepatisation.

  1. Please describe the clinical manifestations in more detail; there is a lot of information in the discussion, but not much in the results that the researchers observed; if fever, how high...; what was the plan of the clinical study

Response: As suggested by the reviewer, we have added some additional photos in Figure 1 during the collection proces. The manifestations of M. ovipneumoniae infection in goats were coughing, fever, labored breathing, mucoid nasal discharge, and gradual emaciation, with a usual temperature of 41°C, which occasionally reached 42 °C. The next step involves conducting a comparative study on treatment options. Given the current breeding situation, achieving herd immunity poses challenges. On one hand, the vaccine price is high, and on the other hand, there exists a weak awareness of immunity among individuals. Therefore, we aim to compare the therapeutic effects of three treatment groups through regression testing in order to formulate an optimal prevention and control plan suitable for the Anhui province.

  1. The volume number should be written in italics

Response: Express gratitude to the reviewers for their meticulous examination. As suggested by the reviewer, the volume number has been italicized.

We tried our best to improve the manuscript and made some changes. Additional minor revisions were made to the manuscrip using track changes to improve English and readability based on suggestions from Elixigen CO. Special thanks for your good comments. We hope that the corrections made will meet with approval.
